# Reuse of Oil Refining Sludge Residue Ash via Alkaline Activation in Matrices of Chamotte or Rice Husk Ash

**DOI:** 10.3390/ma16072801

**Published:** 2023-03-31

**Authors:** Almudena García-Díaz, Salvador Bueno-Rodríguez, Luis Pérez-Villarejo, Dolores Eliche-Quesada

**Affiliations:** 1Department of Chemical, Environmental and Materials Engineering, Higher Polytechnic School of Jaén, University of Jaén, Campus Las Lagunillas s/n, 23071 Jaén, Spain; 2Center for Advanced Studies in Earth Sciences, Energy and Environment (CEACTEMA), University of Jaén, Campus Las Lagunillas s/n, 23071 Jaén, Spain

**Keywords:** oil refining sludge ash, rice husk ash, chamotte, geopolymers, alkaline activated cements, mechanical properties

## Abstract

The aim of this work is to investigate the possibility of reusing ashes obtained by the calcination of industrial sludge from the oil refining industry (ORSA) as a secondary raw material in the manufacture of alkaline activated cements or geopolymers. The incorporation behavior of 5–20 wt.% of residue in binary mixtures with rice husk ash (RHA) or chamotte (CHM) was evaluated. The cements were activated with a sustainable alternative activating solution obtained from NaOH (10 M) and diatomaceous earth. The specimens were cured at room temperature. Physical and mechanical properties were determined, and the reaction products were characterized by X-ray diffraction (XRD), Fourier transform infrared spectroscopy (FTIR) and scanning electron microscopy–energy dispersive X-ray spectroscopy (SEM-EDX). The results indicate that the addition of ORSA (5–20 wt.%) to RHA and CHM improves the mechanical strength of alkaline activated cements with maximum compressive strengths of 30.6 MPa and 15.7 MPa, respectively, after 28 days of curing, with the incorporation of 20 wt.% waste. In these mixtures, the sludge acts as a source of aluminum, promoting the formation of a higher amount of geopolymer gel N-A-S-H in materials using RHA as a precursor and also (N)-(C)-A-S-H gel in cements using CHM.

## 1. Introduction

Portland cement (PC) is the most widely used binding material in modern construction and is one of the most widely produced materials in the world. The global cement industry is looking for experimental ways to develop cements that require less energy in their manufacture, degrade the environment less due to the high consumption of mineral resources and emit less polluting gases into the atmosphere, which contribute almost 7% of global CO_2_ emissions (2.5 Gt/year), aggravating global warming and its consequences [1,2,3]. In addition, the industry is facing challenges such as increasing demand for cement and, at the same time, limited limestone reserves and increasing carbon taxes [4]. The so-called circular economy is emerging as an environmental and economic concept that aims to keep the value of products, materials and resources (water, energy, etc.) in the economy for as long as possible, and to minimize waste generation, advocating the need for a transition from a linear model (extraction, manufacture, use and disposal) to a circular one. Resource efficiency in Europe is thus one of the initiatives that are part of the Horizonte Europa strategy to generate smart, sustainable and inclusive growth [5]. Therefore, from a technical, environmental and economic point of view, improving waste recovery can lead to lower environmental impacts, as well as opening up new markets and jobs and fostering less dependence on imports of raw materials. In addition, there is a need to develop new technological tools to stop the environmental impact of the manufacture of building materials, which can be reduced by using waste or by-products in their formulation. The cement industry has developed alternative cements, such as alkaline activated cements (AACs) or geopolymer cements, the most promising of which are green cements, due to their lower environmental impact [6]. These are non-Portland cements based solely on natural minerals, industrial waste or by-products and an alkaline activator [7]. Two models of alkali-activated bonding systems have been established [8]. In the first alkaline activation model, silicon- and aluminum-rich precursors (metakaolin or class F fly ash) are activated with alkaline solutions, forming during the reaction a matrix in the form of a three-dimensional 3-D aluminosilicate framework, forming a tetrahedral structure. Davidovits called this group geopolymer, as they have a polymeric structure [9,10]. In this group, the geopolymerization reaction results in a geopolymer gel or alkaline aluminosilicate gel binder phase (N-A-S-H) [11]. In the second alkaline activation model, silicon- and calcium-rich precursors (steel slags) are alkaline activated, whose main reaction product is the hydrated calcium aluminosilicate gel (C-A-S-H), similar to the gel generated in the hydration of a normal Portland cement (C-S-H gel) but with lower calcium/silica (C/S) ratios [12]. Therefore, geopolymers or alkaline-activated cements are manufactured with a low-cost technology, which allows the use of waste, reduces energy consumption and reduces carbon dioxide emissions as the synthesis takes place at temperatures close to ambient temperature [13].

Regarding the use of alkaline solutions, the most commonly used are alkaline hydroxides (NaOH, KOH) and commercial sodium silicate solutions (waterglass). However, most of the emissions and energy consumption of geopolymers can be attributed to commercial activators. Sodium silicate is produced from silicon oxide and sodium carbonate, natural resources, at high temperatures (1300 °C), so its production consumes a lot of energy and generates CO_2_ in the atmosphere [14,15]. Thus, supplementary sources of silica are needed to reduce the economic and environmental impact of sodium silicate production, and in that way, rice husk ash, diatomaceous earth or glass can represent an attractive option due to their high silica content to produce silicate solutions by chemical reaction with NaOH or KOH [16,17,18]. Therefore, it is necessary to look for alternative solutions to commercial alkaline silicates to achieve waste-based geopolymers or alkaline-activated materials with a near-zero carbon footprint.

In today’s world, with dwindling natural resources and energy crises, the importance and need to develop a sustainable approach towards environmentally sound waste management cannot be ignored [19]. Rice husk ash (RHA) is an agro-industrial by-product generated during the combustion process of silica-rich rice husk. Chamotte is obtained by crushing fragments of fired bricks that are defective. Firing causes deterioration of its crystalline structure and formation of silica and amorphous alumina [20]. RHA and chamotte have been used in many fields, including the construction industry. Both wastes have been used as a source of aluminosilicates in the manufacture of geopolymers [21,22,23,24,25].

Biosolids or sewage sludge is an important type of waste [26] resulting from different physical and biological wastewater treatment processes [27]. Considering that sludge disposal represents between 25% and 65% of the total management costs of a wastewater treatment plant [28], the valorization of the obtained by-products becomes a necessity within the framework of sustainable development and circular economy. According to data from the National Sludge Register in Spain [29], its final destination has been agricultural use, if its content in heavy metals is adequate, or incineration, although part of these wastes is deposited in controlled landfills. However, environmental constraints are increasing, and so is the quality required of sludge, which is leading to an increase in the complexity of sludge treatment and its cost, and to a search for recovery processes that comply with legislation. Currently, a great effort is being made to find new viable alternatives for the use of this waste, giving rise to some very promising solutions, such as its use as a raw material for construction materials [30,31]. The use of sewage sludge as a secondary raw material in the manufacture of geopolymers has been studied by other authors, generally using metakaolin, steel slag or coal fly ash and commercial alkaline activators as the main precursor [32,33,34,35,36,37].

This study promotes the manufacture of binary alkaline activation cements (AACs) using rice husk ash or chamotte as the main raw material and oil refining sludge residue ash as a secondary raw material using a supplementary source of silica, spent diatomaceous earth combined with 10 M NaOH, instead of commercial sodium silicate for alkaline activation. First, raw materials characterization and alkali-activated materials were performed. Next, physical properties, such as true density, bulk density, apparent porosity, water absorption and total porosity, and mechanical properties, such as flexural and compressive strength, were determined. The microstructure was characterized by X-ray diffraction (XRD), Fourier transform infrared spectroscopy (FTIR) and scanning electron microscopy–energy dispersive X-ray spectroscopy (SEM-EDX). Therefore, the aim is to obtain more environmentally friendly geopolymer cements using waste as a precursor and activator used as a partial substitute for Portland cement.

## 2. Materials and Methods

### 2.1. Raw Materials and Characterization

Rice husk ash (RHA) was supplied by Herba Ricemills S.L., a rice-producing industry in San Juan de Aznalfarache (Seville, Spain). The powders were used after being ground in a ball mill for 10 min at 350 rpm and sieved to a particle size below 100 µm.

The chamotte (CHM) is obtained by crushing or grinding fragments of defective fired ceramic pieces supplied by the company Ladrillos Bailén S.A., located in Bailén (Jaén, Spain). The bricks were crushed in a jaw mill and then in ball mill for 1 h at 350 rpm and sieved to a particle size of less than 100 µm.

The oil refining sludge residue was supplied by the oil and fat refining factory Aceites del Sur Coosur, S.A., located in Vilches (Jaén, Spain). The sludge was oven dried at 100 °C, ball mill ground, sieved to a grain size of 100 µm and calcined at a constant rate of 10 °C/min up to 750 °C (2 h) to obtain the oil refining sludge ash (ORSA).

The preparation of raw materials is carried out according to the flow chart (Figure 1).

The particle size of precursors was determined using a Malvern Mastersizer 2000 (Malvern, UK). The particle size distribution of the precursors is presented in Figure 2. The average particle size of the RHA, CHM and ORSA (D50) is 24.3, 8.0 and 66.9 µm, respectively.

The crystalline phases were identified by X-ray diffraction (XRD) using an Empyrean X-ray machine with a PIXcel-3D detector from PANalytical (Malvern, UK), using Cu K radiation (Kα = 1.5406 Å) in a 2 θ range from 10 to 60, and a step size of 0.02. HighScore software was used to determine the crystalline phases. The XRD pattern of the RHA residue (Figure 3a) shows diffraction peaks corresponding to silica in the form of cristobalite (Ref cod. 96-900-9687) (SiO_2_ polymorph formed during the RHA combustion process) as the main crystalline phase. For the CHM waste (Figure 3b), SiO_2_, quartz (Ref cod. 96-901-2601), hematite, Fe_2_O_3_ (Ref cod. 96-900-0140), illite (Ref cod. 96-900-9666), plagioclase as albite (Ref cod. 96-155-6999) and alkali feldspars (Ref cod. 96-591-0069) are identified as crystalline phases.

For the ORSA residue (Figure 3c), the main crystalline phase is aluminum phosphate (AlPO_4_) (Ref. cod. 96-153-1952). RHA and ORSA residues show a broad amorphous halo at 15–30° 2 θ, typical of the glassy amorphous phase.

The chemical composition of the raw materials was determined by X-ray fluorescence (XRF) using a Philips Magix Pro model PW-2440 instrument (Andover, MA, USA), and the semiquantitative results are given in Table 1.

The RHA contains a high proportion of silica, SiO_2_ (73.6 wt.%), with less than 2 wt.% of (K_2_O, P_2_O_5_, CaO and MgO). The waste has a high LOI content (20.8 wt.%), indicating a high amount of organic matter due to incomplete combustion of the biomass, rice husk [38]. The CHM residue has mainly a high SiO_2_ content (63.1 wt.%), with significant additional contents of Al_2_O_3_ (12.1 wt.%), CaO (8.7 wt.%) as well as Fe_2_O_3_ (4.7 wt.%). The ORSA residue is mainly composed of Al_2_O_3_ (53.6 wt.%) and P_2_O_5_ (27.3 wt.%), with minor contents of Na_2_O (6.3 wt.%) and SiO_2_ (2.1 wt.%).

Figure 4 shows SEM images of the raw materials, RHA, CHM and ORSA. A wide particle size distribution is observed in all precursors, with micrometer-sized particles as well as some larger ones. Most of the RHA and CHM particles are irregular in shape with sharp edges and rough surface texture, with a smaller proportion of rounded particles. In contrast, ORSA has a higher proportion of rounded particles and a lower proportion of angular particles.

### 2.2. Alkali-Activated Cement Preparation and Characterization

#### 2.2.1. Alkaline Solution

The alternative alkaline solution was prepared by dissolving spent diatomaceous earth (89.09% SiO_2_), supplied by the company Heineken (Jaén, Spain), used in the beer filtration process, in a solution of 10 M NaOH [39,40] in a 1:4 ratio. The ratio was selected according to previous tests. The temperature of the solution was kept constant at 80 ± 2 °C by using a thermostatized bath for 6 h to improve the dissolution of silica. Preliminary tests carried out indicate that the solid residue containing non-solubilized materials or impurities gives rise to alkaline-activated cements with reduced mechanical properties. Thus, the generated sodium silicate solution was vacuum filtered and cooled to room temperature. The liquid residue contains dissolved SiO_2_ together with NaOH, resulting in a purer synthetic sodium silicate that gives rise to better compressive strength. The procedure is the one used by other authors [16,18]. Optimal geopolymers were fabricated using a commercial alkaline solution with sodium hydroxide (NaOH) (98% purity, Panreac) and Panreac sodium silicate (Na_2_SiO_3_) solution (29.2% SiO_2_; 8.9% Na_2_O; and 61.9% H_2_O). NaOH pellets were dissolved in distilled water to obtain a 10 M solution; then, once cooled, a sodium silicate was added with a Na_2_SiO_2_/NaOH mass ratio of 1:1, and the solution was stirred and cooled prior to use.

#### 2.2.2. Preparation of Alkaline Activation Cements

The addition of ORSA (5–20 wt.%) as a source of aluminum to silicon-rich sources such as RHA or CHM was studied in order to obtain binary alkaline activation cements.

First, the precursors were mixed in dry state in a Proeti planetary mixer at slow speed (140 ± 5 rpm) for 90 s. The alternative or commercial activator solution was then poured into the mixture and homogenized for 90 s. The mixing was then stopped for 90 s to remove the paste adhering to the inner wall. Finally, the paste was mixed for 90 s at high speed (285 ± 10 rpm) to complete the mixing of the paste. The pastes were poured into stainless steel molds to obtain prismatic samples (1 × 1 × 6 cm^3^). Samples were covered with film and cured at room temperature (20 °C). After 24 h, they were demolded and kept at room temperature (20–25 °C) in a laboratory environment until the day of testing, 7 and 28 days. Details of all prepared samples are given in Table 2. Samples using alternative activating solutions were designated as CHM-xORSA or RHA-xORSA, depending on whether the precursor used is chamotte or rice husk ash, where x indicates the ORSA content added. Samples using commercial activating solution are suffixed with SS (indicating that commercial sodium silicate solution was used).

### 2.3. Characterization of the Alkaline Activation Cements

The true density of the alkaline activated cements was determined by pycnometry using ethanol as solvent. Bulk density, water absorption and apparent porosity were determined according to the Archimedean principle following the UNE-EN 1015 standard [41]. The total porosity of the geopolymers was obtained from the ratio between the bulk density and the true density with Equation (1).
(1)$$\mathrm{Total}\;\mathrm{porosity}\;\left(\%\right)=\left(1-\;\frac{\mathrm{bulk}\;\mathrm{density}}{\mathrm{true}\;\mathrm{density}}\right)\;\times \;1000$$

Flexural and compressive strength was tested according to UNE-EN 1015-11:2000/A1:2007 [42]. An MTS Insight 5 machine (5 kN capacity) with a displacement speed of 1.0 mm/min was used to determine the flexural strength. A universal testing machine, MTS 8101 (100 kN), with a displacement speed of 2 mm/min, was used to determine the compressive strength. Three samples were tested to determine the average value of the flexural strength, and the six halves were used to obtain the average value of the compressive strength.

The mineralogical phases of the alkaline-activated cements were determined by XRD using the same equipment and conditions as for the precursors.

The identification of functional groups was carried out by attenuated total reflectance Fourier transform infrared spectroscopy (ATR-FTIR) using the Vertex 70 Bruker (Billerica, MA, USA) in the range 4000–400 cm^−1^.

The morphology of the alkaline activated cements was observed using scanning electron microscopy (SEM) using a JEAL model SM 840 (Peabody, MA, USA) assisted by energy dispersive X-ray spectroscopy (EDS). The samples were placed on an aluminum grid and carbon coated using the JEOL JFC 1100 sputter coater (Tokyo, Japan).

## 3. Results and Discussion

### 3.1. Physical Properties of Alkaline Activated Cements

Table 3 shows the bulk density, water absorption and apparent porosity as a function of curing time for the different alkaline-activated cements. These properties could not be determined for the RHA-5ORSA binders due to their dissolution in water, indicating that no geopolymer gel is formed in these cements and only N-A-S-H phases are formed by alkaline activation. The bulk density of the geopolymers increases with the addition of ORSA, while water absorption and apparent porosity decrease. For the samples with CHM, the bulk density after 28 days of curing increases from 1299 kg/m^3^ for CHM-5ORSA cements to 1529 kg/m^3^ for CHM-20 ORSA specimens, while water absorption values of 17.02 and 5.55% and bulk porosity values of 22.10 and 8.50%, respectively, were obtained. For the samples using RHA, higher values of bulk density and lower values of water absorption and apparent porosity are obtained, developing a maximum bulk density value of 1677 kg/m^3^ and minimum water absorption and apparent porosity values of 1.86% and 3.1%, respectively, for the RHA-20ORSA geopolymers.

The true density of RHA, CHM and ORSA is 2120 kg/m^3^, 2721 kg/cm^3^ and 2566 kg/m^3^, respectively; therefore, these data indicate the formation of more compact structures when RHA is used as a precursor. The total porosity data after 28 days of curing (Figure 5) indicate a decrease in this property with the addition of ORSA, obtaining similar values of total porosity for the RHA-20ORSA and CHM-20ORSA samples of approximately 17.0%, indicating a higher fraction of closed porosity in the samples using RHA as a precursor. The l/s ratio used for both precursors is similar, with values of 1.55 and 1.45 for geopolymers using RHA and CHM, respectively. The higher porosity of CHM-xORSA cements could be due to a major excess of liquid in the mixture using the CHM precursor, due to the lower amount of water to acquire adequate consistency and workability, that escapes to the environment during the early stages of the curing process [43].

In addition, the higher amount of amorphous structure in the RHA precursor indicates a higher amount of reactive silica in the geopolymerization process, leading to a higher formation of hydration products that fill the pores and increase the density of the alkaline activated cements (AACs). For geopolymers using the commercial activator, a significant decrease in bulk density and a notable increase in bulk porosity and water absorption are observed, due to the higher viscosity of the mixture and rapid hardening that does not allow adequate workability, affecting the quality of compaction and consolidation.

As for the influence of the curing time, an increase in bulk density and a decrease in water absorption and apparent porosity are observed, due to the progress of the geopolymerization reaction that results in more compact hydration products.

### 3.2. Flexural and Compressive Strength

Flexural and compressive strength testing was conducted to evaluate the influence of ORSA content on the mechanical performance of RHA- and CHM-based geopolymers (Figure 6). The results of the mechanical tests indicate that the incorporation of increasing amounts of ORSA to the RHA and CHM precursors using an alternative activating solution produced an increase in these mechanical properties, obtaining optimum flexural strength values of 8.9 MPa and 12.9 MPa and compressive strength of 15.7 MPa and 30.6 MPa, for the CHM-20ORSA and RHA-20ORSA AACs after 28 days of curing, respectively.

The addition of ORSA allows the continuity of the geopolymerization reaction, as well as the multicondensation of the aluminosilicate in the mixture, increasing the degree of geopolymerization [44], resulting in the production of a three-dimensional structure and forming a sodium aluminosilicate hydrate gel (N-A-S-H) in the geopolymers using RHA and CHM as precursor together with the formation of a calcium aluminate gel (N)-(C)-A-S-H) hybrid gel by the partial replacement of the sodium in the N-A-S-H gel due to the calcium content present in the CHM. N-A-S-H and (N)-(C)-A-S-H gel helps to develop a more compact and dense binder structure. Thus, the mechanical performance of the geopolymers increases [44]. The increase in mechanical properties with the addition of ORSA is probably related to the effect of alumina in the degree of geopolymerization and to a filler effect [45]. The flexural and compressive strength is lower in the ACCs using CHM, which could be due to, since alkaline activation phenomena occurred in the materials with the formation of effective binder phases that bind the partially reacted crystalline particles, the lower densification of the samples and higher proportion of open porosity.

The obtained flexural and compressive strength values of the specimens using the commercial solution are very low, due to the rapid hardening of the mixture, which prevents the compaction of the mixture as indicated by the low bulk density values and high total porosity values. This rapid hardening prevents further dissolution of the silico-aluminates minerals due to the rapid formation of oligomer products covering the surface of the precursors in the initial reaction stage, leading to incomplete hydration of the raw materials. The rapid evaporation of water also lowers the strength of the geopolymers [46,47].

In all the pastes, the flexural and compressive strength increases with curing time from 7 to 28 days of curing, indicating a reaction with time. However, this increase is more noticeable in the pastes using RHA as a precursor. The dissolution of RHA at an earlier age was slower compared to the CHM precursor, with its contribution to compressive strength development being more evident at a later age. The early compressive strength, after 7 days of curing in CHM-xORSA geopolymers, reaches approximately 75% of the 28-day compressive strength [48].

The compressive strength values obtained for the RHA-xORSA samples are similar to those estimated from the tensile strength for geopolymers using RHA and aluminum-anodizing sludge in a mixture of sodium silicate and sodium hydroxide as an alkaline solution, with values between 19 and 39.8 MPa, with samples cured at 40 °C [45]. They are far superior to those obtained by other authors using uncalcined urban sewage sludge and RHA with compressive strengths of less than 10 MPa [49]. These results differ from those obtained by Istuque et al., 2019 [32], who incorporated 10 wt.% of sewage sludge ash into metakaolin-based geopolymers using sodium silicate as an activator and different SiO_2_/Na_2_O molar ratios (0.8 and 1.6). The incorporation of 10 wt.% sewage sludge ash results in geopolymers with lower compressive strengths compared to the control geopolymer when a SiO_2_/Na_2_O ratio of 0.8 is used, obtaining similar compressive strengths at 14 days of curing for the geopolymers using a SiO_2_/Na_2_O = 1.6 ratio.

The compressive strength results are similar to those obtained by other authors using as precursor only chamotte activated with a solution of sodium silicate and sodium hydroxide for an activator modulus Ms between 1.2 and 2 and% alkali dosage of 4% [50]. However, they are slightly lower than those obtained by the addition of 30 wt.% of water potabilization sludge to clayey sediments activated with a sodium silicate and a 14 M NaOH solution in a mass ratio of 2.32. Compressive strength of 22.9 MPa after 28 days of curing at room temperature was obtained [51].

### 3.3. XRD of Geopolymers

The XRD patterns of the samples with 10 wt.% and 20 wt.% ORSA and RHA or CHM as precursors after 28 days of curing activated with the alternative solution and with the commercial solution are shown in Figure 7. The results show that the crystalline phases present in the precursors, cristobalite in RHA, quartz, hematite, illite, albite and nephelite in the CHM residue and aluminum phosphate in ORSA, appear in the RHA-xORSA geopolymers. However, some of the residue phases are partially dissolved by the activating solution to provide Si^4+^ and Al^3+^ ions, which participate in the geopolymerization process by forming an aluminosilicate gel [38]. In the geopolymer samples, diffraction peaks not present in the raw materials corresponding to sodium carbonate (Ref. number 96-210-6395) are also observed both for the geopolymers using RHA and those using CHM. In specimens using CHM, diffraction peaks corresponding to calcium carbonate are also observed (Ref. number 96-210-6395), indicating that the excess of sodium in the geopolymers using RHA as a precursor and the excess of sodium and calcium in the geopolymers using CHM residue react with atmospheric CO_2_, producing the carbonation of the binders. After alkaline activation, a hump at 2 θ 20–38° is observed as a diffraction characteristic of the geopolymer gel and is more pronounced in the pastes using RHA, in accordance with the higher amount of reactive amorphous material in this precursor, according to the compressive strength data of the RHA-xORSA geopolymers. The activator causes the breakage of Si-O-Si and Si-O-Al structures and promotes repolymerisation, leading to the formation of amorphous N-A-S-H and (N)-(C)-A-S-H gel by combining the tetrahedra of [SiO_4_] and [AlO_4_] with Na^+^ ions and some Ca^+^ ions. The hybrid gel (N)-(C)-A-S-H is only formed in geopolymers using CHM as a precursor according to the chemical composition of the residue (Table 1), as indicated by the SEM analysis of the geopolymers (see Section 3.5).

### 3.4. FTIR of Geopolymers

The FTIR spectra of the selected RHA-xORSA and CHM-xORSA geopolymers using the alternative activating solution and the commercial solution after 28 days of curing are shown in Figure 8. The FTIR spectra of the precursors are included for comparison. In the FTIR spectra of the ORSA, RHA and CHM residue, an absorption band is identified at approximately 1061, 1063 and 970 cm^−1^, respectively, which is associated with asymmetric stretching vibrations of the T-O-Si bond (T = Al or Si) typical of aluminosilicate species [52]. Another significant absorption band appearing in the three residues around 453, 449 and 446 cm^−1^ is related to O-Si-O bending [53]. In the RHA precursor, bands centered at 792 and 617 cm^−1^ are also observed; the former is due to the stretching vibrations of the Si-O bonds, characteristic of amorphous silica [54]. The band identified at 617 cm^−1^ may be related to the cristobalite phase [55]. In the CHM residue, we observe bands centered at 782 and 763, 681 and 668 and 542 cm^−1^ assigned to the Al-O stretching vibration, Si-O-Si symmetric stretching vibration and Si-O-Al deformation vibration, respectively [56,57]. In addition, centered bands are observed at 1428 and 861 cm^−1^, suggesting the presence of O-C-O bonds of CO_3_^2−^ groups associated with carbonate phases. In CHM and in the ORSA residue, absorption bands centered at 3345–3300 cm^−1^ and 1603 cm^−1^ assigned to O-H stretching and bending vibration are observed [58]. In the geopolymers using RHA and ORSA as a precursor, the band centered at 1063 cm^−1^ in RHA and at 1061 cm^−1^ in ORSA shifted in the range of 953–958 cm^−1^ after the samples reacted with the alternative alkaline solution, and up to 990 cm^−1^ when commercial sodium silicate is used. While in the geopolymers using CHM and ORSA, the band centered at 970 cm^−1^ in the residue shifts to 951–953 cm^−1^ when using the alternative solution and to 965 cm^−1^ when using commercial sodium silicate. When the SiO_4_ tetrahedron is partially substituted by the AlO_4_ tetrahedron during the hydration reaction, there is a change in the local chemical environment of the Si-O bond that explains the shift towards the low wavenumber, being more pronounced in the RHA-xORSA geopolymers. This shift suggests polycondensation of the bonds in the alkaline medium and the formation of an amorphous aluminosilicate gel [59], N-A-S-H gel in RHA-xORSA and N-A-S-H and (N, C)-A-S-H gel in CHM-xORSA geopolymers. An increase in the intensity of this band is observed as the ORSA content increases, which could indicate the formation of more aluminosilicate gels, in agreement with the compressive strength data for the geopolymers using the alternative activating solution. All geopolymers show absorption bands centered at 3150–2900 cm^−1^ and 1610–1620 cm^−1^ assigned to the stretching and bending vibrations of the O-H bonds, indicating that during the geopolymerization reaction, water was absorbed on the surface or trapped in the pores of the geopolymer. Another confirmation of the occurred geopolymerization can be found in the modification of the spectra in the range 600–800 cm^−1^ related to the Al-O-Si and Si-O-Si vibrations that are due to the formation of N-A-S-H and (N, C)-A-S-H gel [60]. In addition, centered bands at 1400–1425 cm^−1^ and 850–860 cm^−1^ are characteristic of CO_3_^2−^ assigned to C-O bending and stretching vibrations, respectively [61,62]. These bands reflect the formation of the sodium carbonate formed by atmospheric carbonation of unreacted NaOH and also CaO in CHM precursor [63], as these bands do not appear in the RHA and ORSA precursors and are less intense in the CHM precursor.

### 3.5. SEM-EDX Analysis of Geopolymers

Scanning electron microscopy (SEM) coupled with energy dispersive X-ray spectroscopy (EDX) was performed on the fracture surfaces of the RHA-10ORSA, RHA-20ORSA and RHA-20ORSA-SS (Figure 9), and CHM-10ORSA, CHM-20ORSA and CHM-20ORSA-SS geopolymers (Figure 10).

In the samples using RHA as a precursor and diatomaceous earth alternative solution as an activator, a dense and homogeneous matrix is observed, which may contribute to the higher compressive strength of these geopolymers. The cohesion of the binder structure is observed, with no unreacted particles and homogeneous gel formation. The alternative activating solution dissolves the reactive SiO_2_ and Al_2_O_3_ species present in the RHA and ORSA precursors, forming N-A-S-H hydrated sodium aluminosilicate gels (Spectrum 1 and Spectrum 2). Most of the observed porosity is closed according to the water absorption data, because most of the pores are filled with gel as soon as the liquid phase is able to reach the precursor particles [64].

In the specimens using CHM as precursor activated with the alternative activating solution, two types of gels were observed, N-A-S-H gel (Spectrum 1) with an elemental composition dominated by silicon, sodium and aluminum, and a mixed (N)-(C)-A-S-H gel dominated by silicon, sodium, calcium and aluminum. Some unreacted and partially reacted CHM particles were also found on the surface (Spectrum 3), due to the higher crystallinity of the precursor, as indicated by XRD data. Closed pores and some cracking can also be observed, especially in the CHM-10ORSA sample. The cracking may occur due to volume changes that occur when forming an amorphous to semi-crystalline (N)-(C)-A-S-H gel within a partially hardened geopolymer gel. The closed porosity may be due to the competition of sodium and calcium for soluble silicates and the space available for their growth; instead of one phase acting as a micro-aggregate to fill the voids, both form two phases of similar size, producing more residual pores, with a consequent reduction in compressive strength [65].

The RHA-20ORSA-SS and CHM-20ORSA-SS binders activated with sodium silicate show a less homogeneous microstructure with less formation of reaction products. A higher amount of unreacted particles is observed due to the rapid hardening of the binders, which does not allow the complete dissolution of the precursors affecting the gel formation and the formation of higher porosity according to the total porosity data. The non-reactive particles, the porosity and the lower proportion of gel formed do not contribute to the mechanical strength of the resulting binders.

## 4. Conclusions

From the research results, the innovative synergistic use of up to 20 wt.% oil refining sludge ash (ORSA) in combination with silica-rich sources, rice husk ash (RHA) or chamotte (CHM) as solid precursors of geopolymer materials can be concluded.

Activation with a viable alternative source to conventional sodium silicate, such as diatomaceous earth, influences the setting time, prolonging it and improving the mechanical and physical properties. The addition of ORSA proved to be very significant in strength development, due to the micro-filling effect of the unreacted particles and partial dissolution of the aluminum that favors gel formation. The RHA-20ORSA geopolymers showed a compressive strength after 28 days of curing at 30.6 MPa, so the material can be a binder comparable to Portland cement. For geopolymers using CHM as a precursor, maximum compressive strength values of 15.7 MPa are reached for CHM-20ORSA specimens.

The RHA residue undergoes a higher degree of geopolymerization under the activation conditions studied, possibly due to the higher amount of reactive amorphous material in the precursor, which results in higher N-A-S-H geopolymer gel formation. The CHM residue undergoes a weaker geopolymerization reaction with the activating solution due to the lower amount of amorphous material in the precursor and the formation of two N-A-S-H and (N)-(C)-S-H gels in similar proportions that compete with each other forming more porous and less resistant materials.

Activation of the binary system RHA-ORSA and CHM-ORSA with a commercial sodium hydroxide (10 M)–sodium silicate solution is not possible. Rapid curing of the blends results in geopolymers with low bulk densities and decreased mechanical properties, with maximum compressive strengths of 5.5 and 2.5 MPa for RHA-20ORSA-SS and CHM-20ORSA-SS, respectively, after 28 days of curing.

These promising results suggest that geopolymers can be obtained using waste as structural and/or non-structural material by adding 20 wt.% ORSA to RHA or CHM using an alternative activating solution from diatomaceous earth as a source of reactive silica, which makes these cements very attractive from an environmental and economic point of view. However, the relatively low mechanical properties of CHM-ORSA cements, the variability in the chemical composition of the raw materials, the need for incineration of oil refining sludge, the uncertainty in long-term performance as well as durability are the main limitations to using this material immediately. Therefore, to determine the potential practical application of the developed geopolymer cements, longer-term studies and durability assessments are needed. Cost and sustainability assessments are also needed to create awareness for the use of waste-based geopolymer cements, which would boost their application.

## Figures and Tables

**Figure 1 materials-16-02801-f001:**
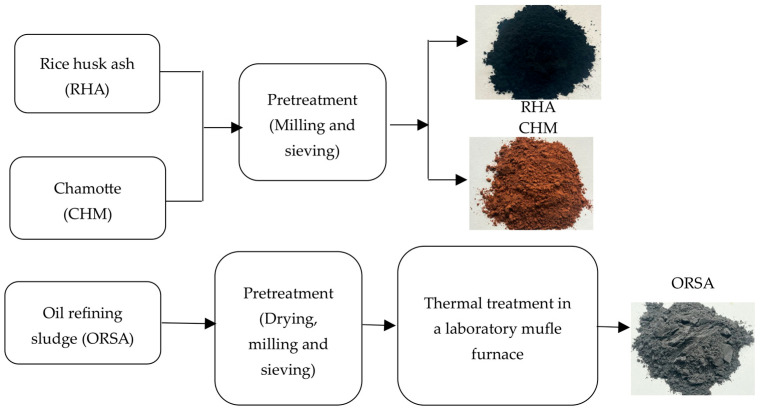
Schematic representation of the preparation of raw materials.

**Figure 2 materials-16-02801-f002:**
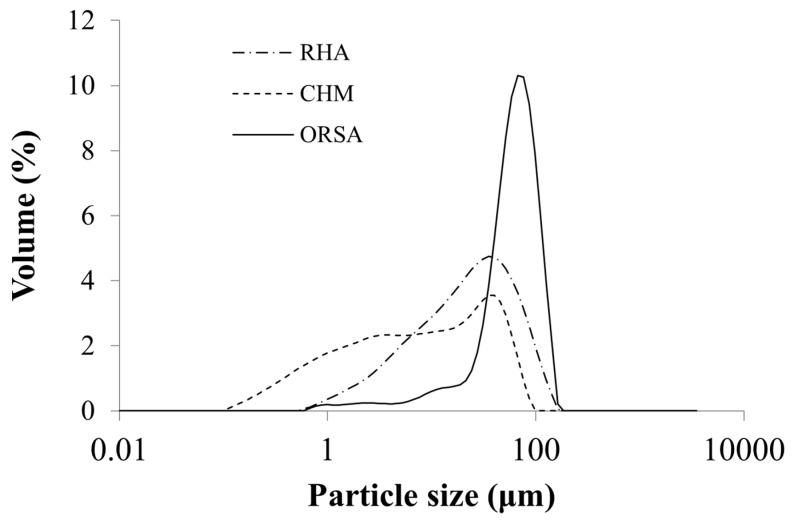
Particle size distribution of raw materials. Rice husk ash (RHA), chamotte (CHM) and oil refining sludge ash (ORSA).

**Figure 3 materials-16-02801-f003:**
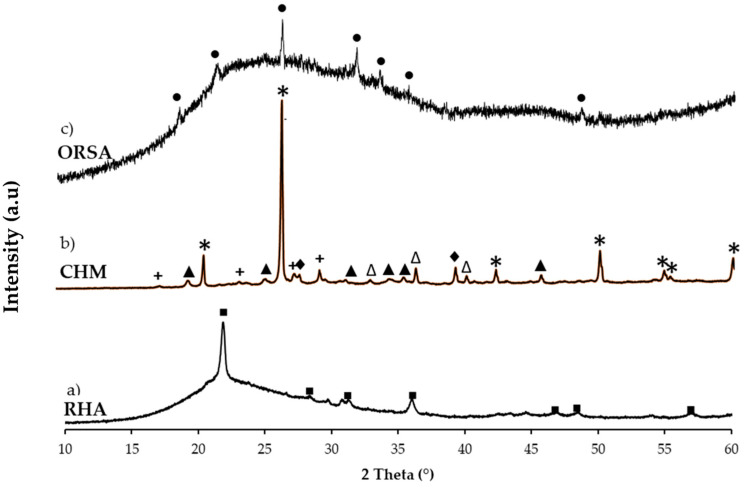
XRD patterns of raw materials: (**a**) Rice husk ash (RHA); (**b**) Chamotte (CHM) and (**c**) oil refining sludge ash (ORSA). (*: quartz; ∆: hematite; +: nephelite; ▲: illite; ♦: albite; ■: cristobalite; ●: AlPO_4_).

**Figure 4 materials-16-02801-f004:**
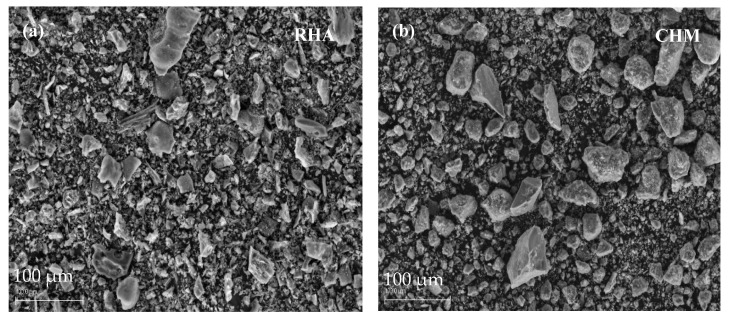
SEM micrographs of raw materials: (**a**) rice husk ash (RHA); (**b**) chamotte (CHM) and (**c**) oil refining sludge ash (ORSA).

**Figure 5 materials-16-02801-f005:**
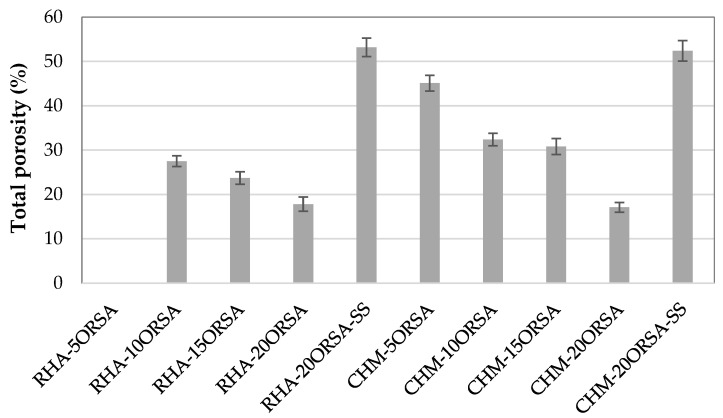
Total porosity of geopolymers at 28 days of curing.

**Figure 6 materials-16-02801-f006:**
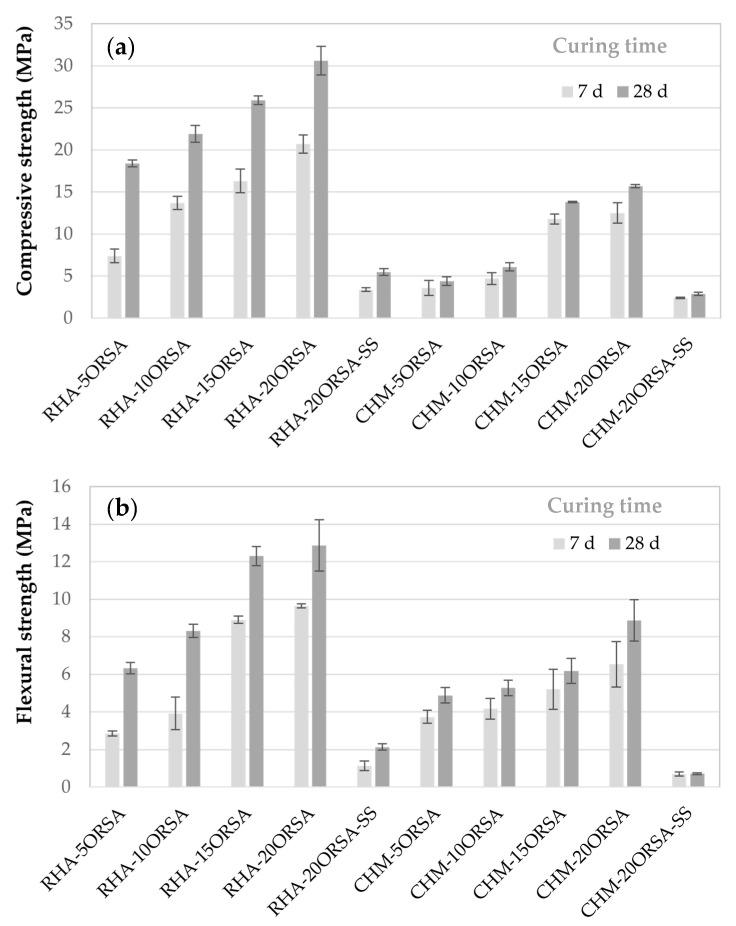
Mechanical properties of geopolymers as function of curing time. (**a**) Flexural strength and (**b**) compressive strength.

**Figure 7 materials-16-02801-f007:**
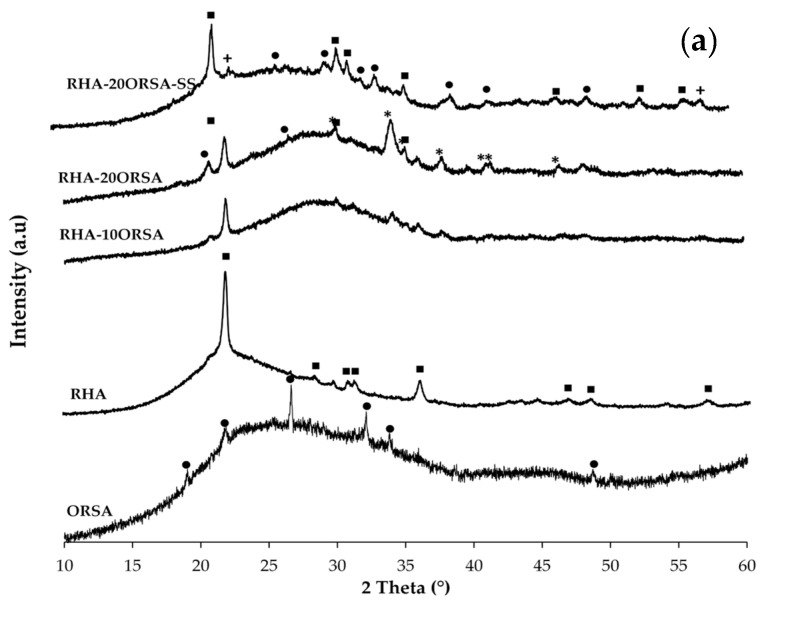
XRD patterns of (**a**) RHA-xORSA geopolymers (+: quartz; * Na_2_(CO_3_); ■: cristobalite; ●: AlPO_4_) and (**b**) CHM-xORSA geopolymers (*: quartz; ∆: hematite; +: nephelite; ▲: illite; ♦: albite; ●: AlPO_4_; ◦ Na_2_(CO_3_); ▫: CaCO_3_).

**Figure 8 materials-16-02801-f008:**
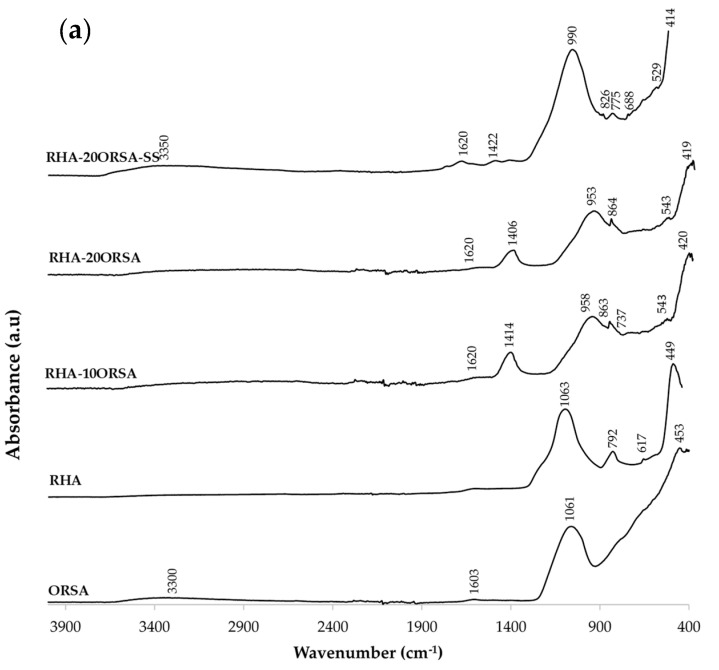
FTIR spectrum of (**a**) RHA-xORSA geopolymers and (**b**) CHM-xORSA geopolymers at 28 days of curing.

**Figure 9 materials-16-02801-f009:**
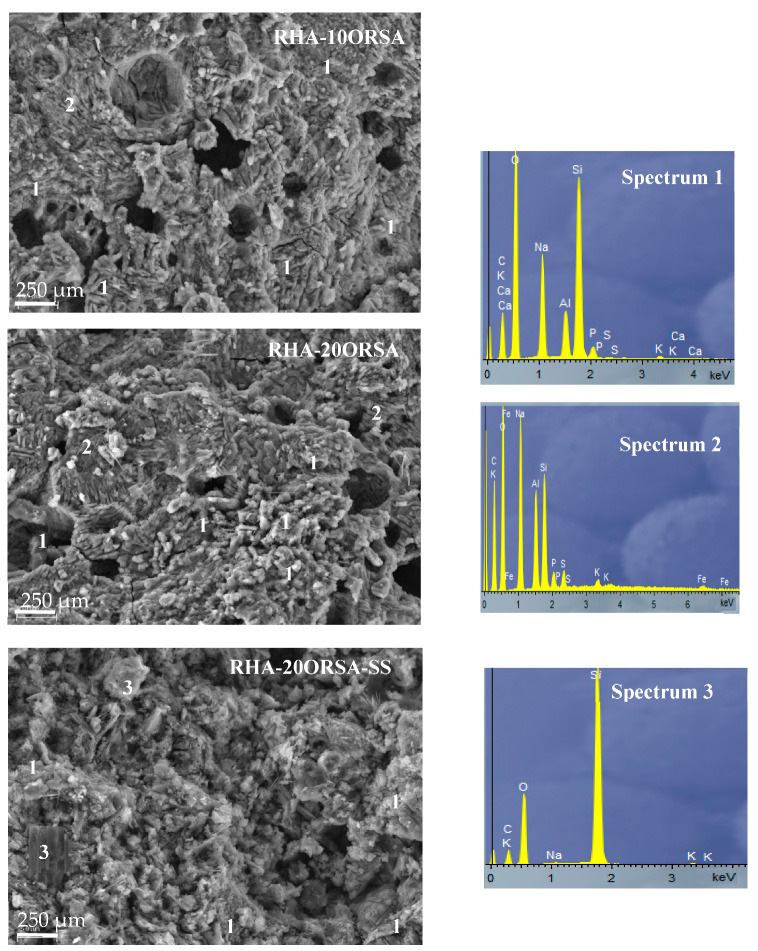
SEM micrographs-EDX analysis of RHA-10ORSA, RHA-20ORSA and RHA-20ORSA-SS geopolymers.

**Figure 10 materials-16-02801-f010:**
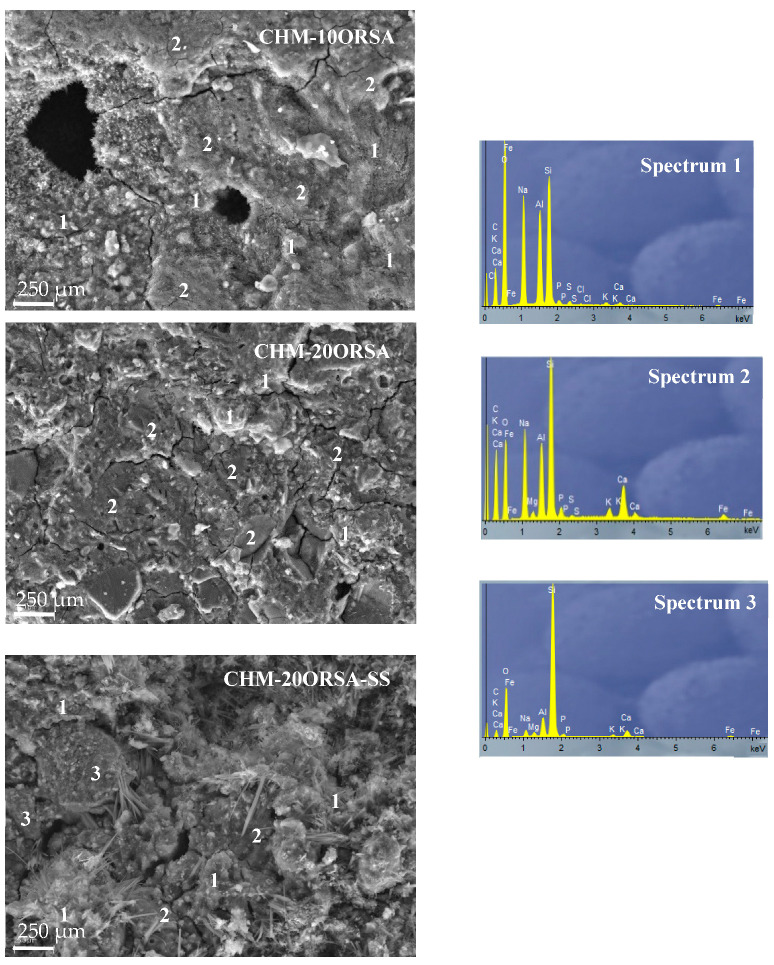
SEM micrographs-EDX analysis of CHM-10ORSA, CHM-20ORSA and CHM-20ORSA-SS.

**Table 1 materials-16-02801-t001:** Chemical composition of raw materials. Rice husk ash (RHA), chamotte (CHM) and oil refining sludge ash (ORSA).

wt.%	SiO_2_	Al_2_O_3_	Fe_2_O_3_	CaO	MgO	Na_2_O	K_2_O	SO_3_	TiO_2_	P_2_O_5_	LOI
RHA	73.6	-	0.29	0.78	0.72	0.15	1.63	0.05	-	1.75	20.8
CHM	63.1	12.11	4.67	8.67	1.88	0.47	3.25	1.18	0.69	0.12	3.60
ORSA	2.11	53.6	0.71	0.76	0.31	6.29	0.36	1.75	0.02	27.3	6.70

**Table 2 materials-16-02801-t002:** Mix compositions of the geopolymers.

Geopolymer	RHA (g)	CHM (g)	ORSA (g)	Alternative Solution (g)	Sodium Silicate (g)	NaOH (g)	H_2_O (g)
RHA-5ORSA	142.5	-	7.5	235.5	-	-	-
RHA-10ORSA	135.0	-	15.0	235.5	-	-	-
RHA-15ORSA	127.5	-	22.5	235.5	-	-	-
RHA-20ORSA	120.0	-	30.0	235.5	-	-	-
RHA-20ORSA-SS	120.0	-	30.0	-	116.25	53.54	87.63
CHM-5ORSA	-	285	15.0	435	-	-	-
CHM-10ORSA	-	270	30.0	435	-	-	-
CHM-15ORSA	-	255	45.0	435	-	-	-
CHM-20ORSA	-	240	60.0	435	-	-	-
CHM-20ORSA-SS	-	240	60.0	-	217.50	53.64	136.96

**Table 3 materials-16-02801-t003:** Bulk density, water absorption and apparent porosity as function of curing time for the different geopolymers.

Geopolymers	Bulk Density (kg/m^3^)		Water Absorption (%)		Apparent Porosity (%)	
Curing Time (Days)	7 d	28 d	7 d	28 d	7 d	28 d
RHA-5ORSA	-	-	-	-	-	-
RHA-10ORSA	1329 ± 34	1670 ± 56	15.2 ± 0.28	5.85 ± 0.65	18.84 ± 0.8	8.90 ± 1.0
RHA-15ORSA	1490 ± 22	1632 ± 10	10.00 ± 0.79	4.09 ± 0.42	14.9 ± 1.0	6.7 ± 0.6
RHA-20ORSA	1594 ± 14	1677 ± 6	5.56 ± 0.44	1.86 ± 0.21	8.9 ± 0.6	3.1 ± 0.3
RHA-20ORSA-SS	932 ± 27	989 ± 28	43.43 ± 3.39	41.03 ± 2.31	40.5 ± 1.9	40.6 ± 1.2
CHM-5ORSA	1233 ± 26	1299 ± 28	20.71 ± 1.16	17.02 ± 1.65	25.6 ± 0.9	22.1 ± 1.7
CHM-10ORSA	1299 ± 28	1481 ± 16	17.02 ± 1.62	10.44 ± 0.82	22.1 ± 1.7	15.5 ± 1.1
CHM-15ORSA	1287 ± 25	1448 ± 15	17.35 ± 1.00	9.14 ± 0.12	22.3 ± 0.9	13.3 ± 0.2
CHM-20ORSA	1515 ± 21	1529 ± 46	8.91 ± 0.81	5.55 ± 0.28	13.5 ± 1.0	8.5 ± 0.5
CHM-20ORSA-SS	1105 ± 03	1119 ± 6	36.82 ± 0.33	38.89 ± 0.41	40.8 ± 0.3	43.6 ± 0.2

## Data Availability

Not applicable.

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
