# Peer review of "Reuse of Oil Refining Sludge Residue Ash via Alkaline Activation in Matrices of Chamotte or Rice Husk Ash"

_materials, 2023, doi:10.3390/ma16072801_

Round 1

Reviewer 1 Report

GENERAL COMMENTS----------------

The papers deal with the mix design of novel AAMs based on different types of solid waste. An alternative activating solution is also proposed. The paper is well written, and the scientific claims are well explained. The authors clearly describe the selection of the materials compositions as well as the characterization methods. Nevertheless, the manuscript contains some laconic parts, as described in the DETAILED COMMENTS. Please check for typing errors.

I suggest minor revision before publication.

DETAILED COMMENTS----------------------------

ABSTRACT

In the sentence: "The incorporation behaviour of (5-20 wt%) of residue in binary mixtures with rice husk ash (RHA) or chamotte (CHM) was evaluated.", please delete the brackets.

To make a clearer list of your precursors, I suggest substituting "diatoms" with: "diatomaceous earth" or "diatomaceous sand".

INTRODUCTION

Refs. 1, and 2 should be updated with most recent papers. Please consider papers: https://doi.org/10.3390/ma13204685; https://doi.org/10.3390/ma15020401; https://doi.org/10.1007/s40030-019-00409-4; https://doi.org/10.1016/j.jclepro.2021.130270; doi: 10.3390/ma14040995.

Lines 49-50: "...due to their 49 lower environmental impact". Please support this statement with a reference paper.

Lines 89-90: "using a supplementary source of silica, spent diatoms, instead of commercial sodium silicate for alkaline activation." spent diatoms alone cannot substitute sodium silicate. Please indicate you are using spent diatoms combined with NaOH.

Line 90, Could authors add a short list of the characterization techniques they are going to present?

2. Materials and Methods

2.1. Raw Materials and characterization

Could you report the unit use in Table 1? wt%? If yes, please check the total amount of oxides plus LOI. I suggest to indicate this chemical composition as semiquantitative. Please check ORSA chemical composition in the Table and in the text. The sum of the oxides should be approx. 90-95 wt%.

2.2. Alkali-activated cements preparation and characterization 

2.2.1. Alkaline solution

Line 205: Is there a real need for filtration? Could you explain the reason why you add this supplementary treatment after dissolution?

2.2.2. Preparation of alkaline activation cements

Line 217: ".......at lente speed...".  Lente????

Line 219: Could you please reformulate the following sentence in order to avoid repetitions?: "Then, the alternative or commercial activating solution was poured into the homogeneous mixture and the mixture was homogenised for 90 seconds."

Line 233: In Table 2, could you check the content of "Alternative activator" for composition: CHM-10ORSA? Is it 43g????

3. Results and Discussion 262

3.1. Physical properties of alkaline activated cements

Line 283: Could you please reformulate the following sentence in order to avoid repetitions?: "The l/s ratio used is very similar with a l/s ratio=1.55 for the geopolymers  using RHA and a l/s ratio=1.45 for the specimens using CHM."

Lines 284-287: Could you please specify why in the CHM-xORSA cements you have excess liquid with respect to other formulations? Is this due to the impervious nature of chamotte particles?

3.2. Flexural and compressive strength

Line 322: Please delete the sentence: "The presence of ORSA, rich in aluminum, increasing the alumina content with respect to silica, increasing the strength of the AACs." This sentence is not as explicative as the following ones and could be misleading for the reader.

3.3. XRD of geopolymers

Line  382: the statement: "The hybrid gel (N)-(C)-A-S-H is only formed in geopolymers using CHM as precursor." has been deduced by the starting chemical compositions of the raw materials or by the position of the amorphous hump in XRD pattern? Please clarify in the manuscript.

3.4. FTIR of geopolymers

Lines 460-462: "These bands reflect the decomposition of the sodium carbonate formed by atmospheric carbonation of unreacted NaOH and also CaO in CHM precursor." Are you sure to use the term: DECOMPOSITION???? Since the peak assigned to carbonates is visible, then I suggest using the term FORMATION.

Line 512: Please check the peaks labels in Fig. 6.

Author Response

Reviewer 1

The papers deal with the mix design of novel AAMs based on different types of solid waste. An alternative activating solution is also proposed. The paper is well written, and the scientific claims are well explained. The authors clearly describe the selection of the materials compositions as well as the characterization methods. Nevertheless, the manuscript contains some laconic parts, as described in the DETAILED COMMENTS. Please check for typing errors.

Thank you very much for your constructive comments, Reviewer 1. These have helped us to greatly improve the original manuscript (changes highligthed in yellow). We reply to all your comments below in bold.     

I suggest minor revision before publication.

DETAILED COMMENTS----------------------------

ABSTRACT

In the sentence: "The incorporation behaviour of (5-20 wt%) of residue in binary mixtures with rice husk ash (RHA) or chamotte (CHM) was evaluated.", please delete the brackets.

To make a clearer list of your precursors, I suggest substituting "diatoms" with: "diatomaceous earth" or "diatomaceous sand".

The brackets have been removed. According to the reviewer diatoms has been replaced by diatomaceous earth.

INTRODUCTION

Refs. 1, and 2 should be updated with most recent papers. Please consider papers: https://doi.org/10.3390/ma13204685; https://doi.org/10.3390/ma15020401; https://doi.org/10.1007/s40030-019-00409-4; https://doi.org/10.1016/j.jclepro.2021.130270; doi: 10.3390/ma14040995.

Thank you, references 1 and 2 have been replaced by the current references indicated by the reviewer.

Lines 49-50: "...due to their lower environmental impact". Please support this statement with a reference paper.

Reference [6] has been added. N. Akhtar, T. Ahmad, D. Husain, A. Majdi, Md. T. Alam, N. Husain, A. K.S. Wayal. Ecological footprint and economic assessment of conventional and geopolymer concrete for sustainable construction. J. Clean. Prod. 2020, 380, 134910. https://doi.org/10.1016/j.jclepro.2022.134910

Lines 89-90: "using a supplementary source of silica, spent diatoms, instead of commercial sodium silicate for alkaline activation." spent diatoms alone cannot substitute sodium silicate. Please indicate you are using spent diatoms combined with NaOH.

Thank you for your appreciation. Done

Line 90, Could authors add a short list of the characterization techniques they are going to present?.

Thank you.  In the last paragraph of the Introduction section a list of techniques used for raw materials characterization and study the materials elaborated has been included

  1. Materials and Methods

2.1. Raw Materials and characterization

Could you report the unit use in Table 1? wt%? If yes, please check the total amount of oxides plus LOI. I suggest to indicate this chemical composition as semiquantitative. Please check ORSA chemical composition in the Table and in the text. The sum of the oxides should be approx. 90-95 wt%.

In Table 1 the oxides are indeed expressed in wt% in oxides. The table and text have been corrected by adding all the components and establishing the correct values. An erroneous composition had been indicated in the text.

2.2. Alkali-activated cements preparation and characterization

2.2.1. Alkaline solution

Line 205: Is there a real need for filtration? Could you explain the reason why you add this supplementary treatment after dissolution?.

Preliminary tests carried out indicate that the solid residue containing non-solubilised materials or impurities gives rise to alkaline activated cements with reduced mechanical properties. For this reason, filtration was carried out, as the liquid residue contains dissolved SiO2 together with NaOH, resulting in a purer synthetic sodium silicate that gives rise to better compressive strength. The procedure is the one used by other authors (H. K. Tchakouté , C. H. Rüscher , S. Kong, E. Kamseu, Cr. Leonelli, Geopolymer binders from metakaolin using sodium waterglass from waste glass and rice husk ash as alternative activators: A comparative study. Construction and Building Materials 114 (2016) 276–289. http://dx.doi.org/10.1016/j.conbuildmat.2016.03.184; M. Torres-Carrasco, F. Puertas. Waste glass as a precursor in alkaline activation: Chemical process and hydration products. Construction and Building Materials 139 (2017) 342-354. https://doi.org/10.1016/j.conbuildmat.2017.02.071). Both references have been incorporated into the manuscript.

2.2.2. Preparation of alkaline activation cements

Line 217: ".......at lente speed...".  Lente????

Sorry, lente is slow. Corrected.

Line 219: Could you please reformulate the following sentence in order to avoid repetitions?: "Then, the alternative or commercial activating solution was poured into the homogeneous mixture and the mixture was homogenised for 90 seconds."

Thank you for the correction, the sentence has been rephrased.

Line 233: In Table 2, could you check the content of "Alternative activator" for composition: CHM-10ORSA? Is it 43g????

It is 435 g,. this has been corrected.Thank you

  1. Results and Discussion

3.1. Physical properties of alkaline activated cements

Line 283: Could you please reformulate the following sentence in order to avoid repetitions?: "The l/s ratio used is very similar with a l/s ratio=1.55 for the geopolymers  using RHA and a l/s ratio=1.45 for the specimens using CHM."

The sentence has been corrected as follows:

The l/s ratio used for both precursors is similar, with values of 1.55 and 1.45 for geopolymers using RHA and CHM, respectively.

Lines 284-287: Could you please specify why in the CHM-xORSA cements you have excess liquid with respect to other formulations? Is this due to the impervious nature of chamotte particles?

Indeed, the CHM residue needs less water to achieve the right consistency and workability than the RHA residue.   A new sentence has been added. “due to the lower amount of water to acquire adequate consistency and workability”.

3.2. Flexural and compressive strength

Line 322: Please delete the sentence: "The presence of ORSA, rich in aluminum, increasing the alumina content with respect to silica, increasing the strength of the AACs." This sentence is not as explicative as the following ones and could be misleading for the reader.

Thank you. This sentence han been deleted.

3.3. XRD of geopolymers

Line  382: the statement: "The hybrid gel (N)-(C)-A-S-H is only formed in geopolymers using CHM as precursor." has been deduced by the starting chemical compositions of the raw materials or by the position of the amorphous hump in XRD pattern? Please clarify in the manuscript.

The presence of this hybrid gel has been deduced, as indicated by the reviewer, from the chemical composition of the raw materials, and subsequently identified by EDS analysis and micrographs. In order to make it more comprehensible for the reader, the sentece has been completed as follows: according to the chemical composition of the residue (Table 1), as indicated by the SEM analysis of the geopolymers (see section 3.5.).

3.4. FTIR of geopolymers

Lines 460-462: "These bands reflect the decomposition of the sodium carbonate formed by atmospheric carbonation of unreacted NaOH and also CaO in CHM precursor." Are you sure to use the term: DECOMPOSITION???? Since the peak assigned to carbonates is visible, then I suggest using the term FORMATION.

 Thank you, the term formation is more appropriate.

Line 512: Please check the peaks labels in Fig. 6.

Peaks labels in Fig. 6 have been checked.

The authors acknolwledge all these reviewer´s comments to improve the presentation and content of the original submitted paper.

Reviewer 2 Report

materials-2269536

Title: Reuse of oil refining sludge residue-ash via alkaline activation in matrices of chamotte or rise husk ash

Reviewer Comments: This article needs major revision

1.      In the entire manuscript, the author has used alkaline-activated cements, but all the results and discussion are related to concrete. Why is that?

2.      Please include a summary of the various sections in the last paragraph of the introduction.

3.      The new section on geopolymerization should be introduced after the introduction as it is a core part of geopolymerization. How could the author miss it? Here are a few articles for the author's reference, which can also be included: doi.org/10.1016/j.conbuildmat.2023.130688; doi.org/10.3390/ma16052044; doi.org/10.3390/polym14183765

4.      In section 2.1, it is unclear whether the author used refining sludge residue directly in the geopolymer or if an intermediate process was followed. The entire process is missing, so please provide more information and include a flow chart if possible.

5.      In Figure 2, it is not clear how to identify that (a) is RHA, (b) is CHM, and (c) is ORSA. Please provide more information or labels to help readers understand their characteristics.

6.      In sections 2.2.1 and 2.2.2, it is unclear how the authors fixed the 10M and 1:4 ratio. Additionally, information on curing conditions, room temperature, and particle size distribution is missing. Please provide more details on these aspects.

7.      In Table 2, it is important to include information on the inert material. Please provide details on its composition and any effects it may have on the results.

8.      The results in Figures 3 and 4 appear to be inconsistent or non-uniform. Please provide an explanation or possible reasons for these discrepancies.

9.      In section 3.2, the discussion is lacking in depth and support from relevant literature. Please provide more elaborate discussions and support them with current research to strengthen the results.

10.  In lines 460 to 462, the author stated that "bands reflect the decomposition of the sodium carbonate formed by atmospheric carbonation of unreacted NaOH and also CaO in CHM precursor." Please provide further explanation or evidence to support this statement.

11.  To improve the clarity of the conclusion, please separate the discussion of microstructure analysis and mechanical results into distinct paragraphs. This will help readers better understand and interpret the findings.

12.  Please ensure that recent and relevant research is cited in the references section to support the arguments and findings presented in the paper.

Author Response

Reviewer 2

Title: Reuse of oil refining sludge residue-ash via alkaline activation in matrices of chamotte or rise husk ash

Reviewer Comments: This article needs major revisión

Thank you very much for your constructive comments, Reviewer 2. These have helped us to greatly improve the original manuscript (changes highlighted in yellow). The authors reply to all your comments below in bold.

  1. In the entire manuscript, the author has used alkaline-activated cements, but all the results and discussion are related to concrete. Why is that?

In the manuscript, alkaline activated cements or geopolymers are always mentioned and discussed. These cements could be used in the manufacture of geopolymer concretes in the same way as Portland cement concretes. However, the manuscript indicates the manufacture and characterization of cements.

  1. 2. Please include a summary of the various sections in the last paragraph of the introduction.

Thank you. A new sentence has been added: First, raw materials characterization and alkali activated materials were performed. Next, physical properties such as true density, bulk density, apparent porosity, water absorption and total porosity and mechanical properties such as flexural and compressive strength have been determined. The microstructure has been characterized by X-ray diffraction (XRD), Fourier transform infrared spectroscopy (FTIR) and Scanning Electron Microscopy - Energy Dispersive X-ray spectroscopy (SEM-EDX).

  1. 3. The new section on geopolymerization should be introduced after the introduction as it is a core part of geopolymerization. How could the author miss it? Here are a few articles for the author's reference, which can also be included: doi.org/10.1016/j.conbuildmat.2023.130688; doi.org/10.3390/ma16052044; doi.org/10.3390/polym14183765

According to the reviewer, a new paragraph has been added to the introduction including some of the recommended references.

  1. In section 2.1, it is unclear whether the author used refining sludge residue directly in the geopolymer or if an intermediate process was followed. The entire process is missing, so please provide more information and include a flow chart if possible.

A Flow chart has been included.

  1. In Figure 2, it is not clear how to identify that (a) is RHA, (b) is CHM, and (c) is ORSA. Please provide more information or labels to help readers understand their characteristics.

Thank you. Done

  1. In sections 2.2.1 and 2.2.2, it is unclear how the authors fixed the 10M and 1:4 ratio. Additionally, information on curing conditions, room temperature, and particle size distribution is missing. Please provide more details on these aspects.

A 10 M NaOH solution was used according to the literature [L Handayani, S Aprilia, Abdullah , C Rahmawati, A M Mustafa Al Bakri, I H Aziz, E A Azimi. Synthesis of Sodium Silicate from Rice Husk Ash as an Activator to Produce Epoxy-Geopolymer Cement. Journal of Physics: Conference Series 1845 (2021) 012072. doi:10.1088/1742-6596/1845/1/012072; M. Torres-Carrasco, F. Puertas. Waste glass in the geopolymer preparation. Mechanical and microstructural characterisation. Journal of Cleaner Production 90, 1 March 2015, Pages 397-408. https://doi.org/10.1016/j.jclepro.2014.11.074]. References have been incorporated into the manuscript.

As for the 1:4 ratio, it was selected after previous tests with activation of other precursors (this information is in a manuscript under writing for publication). It shows thatdifferent amounts of diatomaceous earth were dissolved in 10 M NaOH solution, obtaining the best mechanical properties for the solution used in this study.

The curing process is indicated in the manuscript.: “Samples were covered with film and cured at room temperatura (20 ºC). After 24 hours, they were demoulded and kept at room temperatura (20-25 ºC) in a laboratory environment until the day of testing, 7 and 28 days”.

The particle size distribution of the precursors has been added.

  1. In Table 2, it is important to include information on the inert material. Please provide details on its composition and any effects it may have on the results.

We really don't  understand the comment, the authors don't know whether by inert material you mean the use of sand. No inert material (aggregate) has been used in this study. The materials manufactured are pastes and the composition is as stated in Table 2.

  1. The results in Figures 3 and 4 appear to be inconsistent or non-uniform. Please provide an explanation or possible reasons for these discrepancies.

Thank you for the comment.Sorry, indeed the bending strength data for RHA and CHM are wrong. Precursor data were changed. Figure 3 has been modified.

  1. 9. In section 3.2, the discussion is lacking in depth and support from relevant literature. Please provide more elaborate discussions and support them with current research to strengthen the results.

Section 3.2. has been revised, improved and supported by recent research.

  1. In lines 460 to 462, the author stated that "bands reflect the decomposition of the sodium carbonate formed by atmospheric carbonation of unreacted NaOH and also CaO in CHM precursor." Please provide further explanation or evidence to support this statement.

According to the reviewer a new reference has been added. On the other hand, as can be seen in the FTIR results of the precursors, except for CHM, no carbonate formation bands are observed. In the CHM precursor, the intensity of this band is much lower than in the alkaline activated cements. A new sentence has been added:

as these bands do not appear in the RHA and ORSA precursors and are less intense in the CHM precursor.

  1. To improve the clarity of the conclusion, please separate the discussion of microstructure analysis and mechanical results into distinct paragraphs. This will help readers better understand and interpret the findings.

Thank you. Done

  1. Please ensure that recent and relevant research is cited in the references section to support the arguments and findings presented in the paper.

Recent and relevant research is cited in the text.

The authors acknowledge all these reviewer’s comments to improve the presentation and content of the original submitted paper.

Reviewer 3 Report

The authors have submitted a well prepared paper on the interesting topic of the Reuse of oil refining sludge residue-ash via alkaline activation in matrices of chamotte or rice husk ash. The paper is clearly presented and provides interesting results. This study is valuable for the practical engineering. However, the following comments are provided to assist the authors to improve the paper:

1) The title, the word “rise husk ash”, is spelt wrong, should be “rice husk ash”.

2) The article's purpose should be clarified in detail, why this study could be beneficial, and a more in-depth conclusion in applications should be provided.

3) Kindly provide the images of rice husk ash (RHA), chamotte (CHM) and oil refining sludge ash (ORSA) used in the present study.

4) Why did the author use a NaOH concentration of 10 molars? Please explain.

5) I would recommend expanding to ref [12-15] the following articles related to RHA geopolymers: doi.org/10.3390/su14169856

6) Conclusions: the author should further explain this research's construction application limitations. Please describe in conclusion.

7) Please propose some improvements and directions for future research.

Thank you for considering my opinion. I encourage the authors to keep on working to improve the revised manuscript.

Author Response

Reviewer 3

The authors have submitted a well prepared paper on the interesting topic of the Reuse of oil refining sludge residue-ash via alkaline activation in matrices of chamotte or rice husk ash. The paper is clearly presented and provides interesting results. This study is valuable for the practical engineering. However, the following comments are provided to assist the authors to improve the paper:

Thank you very much for your constructive comments, Reviewer 3. These have helped us to greatly improve the original manuscript (changeshighlightedin yellow). We reply to all your comments below in bold.

1)         The title, the word “rise husk ash”, is spelt wrong, should be “rice husk ash”.

Thank you. Corrected.

2)         The article's purpose should be clarified in detail, why this study could be beneficial, and a more in-depth conclusion in applications should be provided.

The purpose of the article has been clarified and a new sentence has been added to the introduction: Therefore, the aim is to obtain more environmentally friendly geopolymer cements using as a precursor and activator waste generated by different industries, used as a partial substitute for Portland cement.

3)         Kindly provide the images of rice husk ash (RHA), chamotte (CHM) and oil refining sludge ash (ORSA) used in the present study.

Thank you. The images have been included as part of the new flow chart (Figure 1). 4)       Why did the author use a NaOH concentration of 10 molars? Please explain.

A 10 M NaOH solution was used according to the literature [L Handayani, S Aprilia, Abdullah , C Rahmawati, A M Mustafa Al Bakri, I H Aziz, E A Azimi. Synthesis of Sodium Silicate from Rice Husk Ash as an Activator to Produce Epoxy-Geopolymer Cement. Journal of Physics: Conference Series 1845 (2021) 012072. doi:10.1088/1742-6596/1845/1/012072; M. Torres-Carrasco, F. Puertas. Waste glass in the geopolymer preparation. Mechanical and microstructural characterisation. Journal of Cleaner Production 90, 1 March 2015, Pages 397-408. https://doi.org/10.1016/j.jclepro.2014.11.074]. References have been incorporated into the manuscript.

5)         I would recommend expanding to ref [12-15] the following articles related to RHA geopolymers: doi.org/10.3390/su14169856.

Thank you, the reference has been incorporated.

6)         Conclusions: the author should further explain this research's construction application limitations. Please describe in conclusion.

Limitations have been incorporated in conclusions

7)         Please propose some improvements and directions for future research.

Future research has been added.

 The authors acknowledge all these reviewer’s comments to improve the presentation and content of the original submitted paper.

Round 2

Reviewer 2 Report

the revised manuscript can be accepted. 

Author Response

Thank you, very much. The authors acknowledge all these reviewer’s comments to improve the presentation and content of the original submitted paper.

Reviewer 3 Report

Thank you for the revised version. The authors have addressed all the concerns of the reviewer and the manuscript can be accepted for publication.

Author Response

Thank you. The authors acknowledge all these reviewer’s comments to improve the presentation and content of the original submitted paper.